

# Beyond RuBisCO: convergent molecular evolution of multiple chloroplast genes in C$_4$ plants

Claudio Casola[1,2] and Jingjia Li[1]

[1] Department of Ecology and Conservation Biology, Texas A&M University, College Station, TX, United States of America

[2] Interdisciplinary Graduate Program in Ecology and Evolutionary Biology, Texas A&M University, College Station, TX, United States of America

## ABSTRACT

**Background**. The recurrent evolution of the C$_4$ photosynthetic pathway in angiosperms represents one of the most extraordinary examples of convergent evolution of a complex trait. Comparative genomic analyses have unveiled some of the molecular changes associated with the C$_4$ pathway. For instance, several key enzymes involved in the transition from C$_3$ to C$_4$ photosynthesis have been found to share convergent amino acid replacements along C$_4$ lineages. However, the extent of convergent replacements potentially associated with the emergence of C$_4$ plants remains to be fully assessed. Here, we conducted an organelle-wide analysis to determine if convergent evolution occurred in multiple chloroplast proteins beside the well-known case of the large RuBisCO subunit encoded by the chloroplast gene *rbcL*.

**Methods**. Our study was based on the comparative analysis of 43 C$_4$ and 21 C$_3$ grass species belonging to the PACMAD clade, a focal taxonomic group in many investigations of C$_4$ evolution. We first used protein sequences of 67 orthologous chloroplast genes to build an accurate phylogeny of these species. Then, we inferred amino acid replacements along 13 C$_4$ lineages and 9 C$_3$ lineages using reconstructed protein sequences of their reference branches, corresponding to the branches containing the most recent common ancestors of C$_4$-only clades and C$_3$-only clades. Pairwise comparisons between reference branches allowed us to identify both convergent and non-convergent amino acid replacements between C$_4$:C$_4$, C$_3$:C$_3$ and C$_3$:C$_4$ lineages.

**Results**. The reconstructed phylogenetic tree of 64 PACMAD grasses was characterized by strong supports in all nodes used for analyses of convergence. We identified 217 convergent replacements and 201 non-convergent replacements in 45/67 chloroplast proteins in both C$_4$ and C$_3$ reference branches. C$_4$:C$_4$ branches showed higher levels of convergent replacements than C$_3$:C$_3$ and C$_3$:C$_4$ branches. Furthermore, we found that more proteins shared unique convergent replacements in C$_4$ lineages, with both RbcL and RpoC1 (the RNA polymerase beta' subunit 1) showing a significantly higher convergent/non-convergent replacements ratio in C$_4$ branches. Notably, more C$_4$:C$_4$ reference branches showed higher numbers of convergent *vs.* non-convergent replacements than C$_3$:C$_3$ and C$_3$:C$_4$ branches. Our results suggest that, in the PACMAD clade, C$_4$ grasses experienced higher levels of molecular convergence than C$_3$ species across multiple chloroplast genes. These findings have important implications for our understanding of the evolution of the C$_4$ photosynthesis pathway.

Corresponding author
Claudio Casola, ccasola@tamu.edu

# INTRODUCTION

Convergent evolution represents the independent acquisition of similar phenotypic traits in phylogenetically distant organisms. Understanding the genomic changes underlying the recurrent emergence of phenotypes is a major goal of molecular evolution. The rapidly increasing taxonomic breadth of genomic resources combined with the development of rigorous frameworks to comparatively investigate molecular changes has accelerated the pace of discovery in this area. For instance, substitutions in coding regions of conserved genes have been implicated in phenotypic changes responsible for adaptation of marine mammals to an aquatic lifestyle (*Foote et al., 2015*; *Zhou, Seim & Gladyshev, 2015*). Other examples of convergent phenotypes whose molecular underpinnings have been investigated include adaptations in snake and agamid lizard mitochondria (*Castoe et al., 2009*), echolocation in mammals (*Parker et al., 2013*; *Thomas & Hahn, 2015*; *Zou & Zhang, 2015*; *Storz, 2016*), and hemoglobin function in birds (*Natarajan et al., 2016*). Convergent traits can evolve *via* changes toward the same derived state (similar phenotype) from the same initial state, which is known as parallelism, or through changes of different initial states, referred to as convergence (*Zhang & Kumar, 1997*; *Storz, 2016*). For the sake of simplification, we will refer to these two processes using the general terminology 'convergence' and 'convergent replacements' throughout the manuscript, unless differently stated.

Several traits are also known to have convergently evolved in land plants (*Li et al., 2018*; *Lu et al., 2018*; *Preite et al., 2019*). One of the most notable examples is represented by the repeated evolution of the $C_4$ photosynthetic pathway in flowering plants. The $C_4$ pathway is a complex functional adaptation that allows for better photosynthesis efficiency under certain environmental conditions, such as dry and warm climates, high light intensity, low $CO_2$ concentration, and limited availability of nutrients (*Knapp & Medina, 1999*; *Long, 1999*). The $C_4$ pathway involves cytological, anatomical and metabolic modifications thought to have evolved multiple times independently in various lineages from the $C_3$ type (*Kellogg, 1999*; *Sage, 2004*; *Sage, Christin & Edwards, 2011*). According to phylogenetic, anatomical and biochemical evidence, the few slightly different variants of the $C_4$ photosynthesis type evolved more than 60 times in angiosperms (*Sage, Sage & Kocacinar, 2012*; *Heyduk et al., 2019*). In grasses (family Poaceae) alone, the $C_4$ pathway has evolved independently ~20 times (*Grass Phylogeny Working Group II, 2012*).

Transitions from $C_3$ to $C_4$ plants resulted from genetic changes that include nonsynonymous substitutions, gene duplications and gene expression alterations (*Christin et al., 2007*; *Christin et al. 2013a*; *Christin et al., 2015*; *Goolsby et al., 2018*; *Heyduk et al., 2019*). It has been suggested that the evolution of the $C_4$ pathways proceeded throughout a series of evolutionary steps wherein the Kranz leaf anatomy typical of this pathway originated first, followed by changes in the expression patterns of key genes and finally by

adaptive modifications of protein sequences (*Sage, Sage & Kocacinar, 2012*; *Christin et al. 2013b*; *Williams et al., 2013*). A model of the adaptive steps leading to $C_4$ photosynthesis showed that key biochemical components of this pathway evolved modularly along a trajectory that was likely very similar across lineages with $C_3$ to $C_4$ transitions (*Heckmann et al., 2013*). Overall, these scenarios suggest that enzymes involved in $C_3$ to $C_4$ transitions experienced similar selective pressures that resulted in the convergent evolution of the same amino acid replacements across $C_4$ lineages.

Evidence of convergent changes in proteins associated with photosynthetic processes has steadily accumulated since genomic data from multiple $C_4$ lineages have become available in the past couple of decades. Most of these studies have focused on the ribulose-1,5-bisphosphate carboxylase/oxygenase (RuBisCO), a large multimeric enzyme that catalyzes the carboxylation of ribulose-1,5-bisphosphate (RuBP), allowing plants to fix atmospheric carbon (*Andersson & Backlund, 2008*). RuBisCO also catalyzes oxygenation of RuBP, which leads to loss of carbon in the process of photorespiration (*Andersson & Backlund, 2008*; *Maurino & Peterhansel, 2010*). RuBisCO's limited ability to discriminate between $CO_2$ and $O_2$ has been attributed to the much higher $CO_2$ to $O_2$ atmospheric partial pressure until ~400 million years ago (*Sage, 1999*; *Sage, 2004*; *Sage, Sage & Kocacinar, 2012*).

Previous studies have revealed multiple convergent amino acid replacements in the large RuBisCO subunit in $C_4$ lineages, encoded by the chloroplast gene *rbcL* (*Kapralov & Filatov, 2007*; *Christin et al., 2008*; *Kapralov et al., 2011*; *Kapralov, Smith & Filatov, 2012*; *Piot et al., 2018*). Some of these convergent replacements have been associated to positive selection of the corresponding codons in $C_4$ monocot and eudicot lineages (*Kapralov & Filatov, 2007*; *Christin et al., 2008*; *Kapralov, Smith & Filatov, 2012*; *Piot et al., 2018*). Notably, biochemical analyses have demonstrated that some recurrent amino acid changes in the large RuBisCO subunit of $C_4$ plants critically alter the kinetics of RuBisCO, resulting in an accelerated rate of $CO_2$ fixation at the beginning of the Calvin-Benson cycle (*Studer et al., 2014*; *Bouvier et al., 2021*). Convergent amino acid changes have also been described in enzymes that are encoded by nuclear genes and play a primary role in the $C_4$ pathway, including the phosphoenolpyruvate carboxylase PEPC (*Christin et al., 2007*; *Besnard et al., 2009*), the NADP-malic enzymes NADP-me (*Christin et al. 2009b*), the phosphoenolpyruvate carboxykinase PEPCK (*Christin et al. 2009a*) and the small RuBisCO subunit (*Kapralov et al., 2011*).

Given the number of biochemical, physiological and anatomical traits that were affected in each evolutionary transition from $C_3$ to $C_4$ photosynthesis (*Heyduk et al., 2019*), it is likely that many genes experienced analogous selective pressures across taxa that include $C_4$ plants. This has been shown to be the case by *Huang et al. (2017)*, who have developed an approach to identify potential genes involved in the transition to $C_4$ photosynthesis using a genome-wide scan for selection along a phylogeny of PACMAD grasses. Of the 88 genes showing signatures of positive or relaxed selection in $C_4$ species, several were not previously known to have a role in $C_4$ photosynthesis. Although this study did not focus on finding convergent replacements, it provided a comprehensive strategy and statistical testing framework to identify novel genes that have likely played a role in the evolution of
$C_4$ grasses. It is possible that a significant fraction of these genes accumulated convergent amino acid replacements during $C_3$-to-$C_4$ transitions.

Another recent, important work has produced the first analysis of convergent replacements across multiple proteins involved in the metabolism of $C_4$ and crassulacean acid metabolism (CAM) among species belonging to the portullugo clade (Caryophyllales). Goolsby and colleagues (*2018*) compared evolutionary patterns in 19 gene families with critical roles in metabolic pathways of both $C_4$ and CAM plants, also known as carbon-concentration mechanisms (CCMs) genes, and in 64 non-CCM gene families. They found convergent replacements in proteins from $C_4$ and CAM lineages, as well as higher levels of convergent replacements in CCM *vs.* non-CCM gene families (*Goolsby et al., 2018*). Additionally, several amino acid replacements that are prevalent among $C_4$ and CAM taxa compared to $C_3$ lineages were identified in this study (*Goolsby et al., 2018*).

Altogether, the results of this and other studies demonstrated that convergent molecular evolution occurred across multiple genes in both $C_4$ and CAM groups. While significant progress has been made towards the detection of signatures of selection associated to the evolution of CCMs (*Huang et al., 2017*; *Piot et al., 2018*), a rigorous framework to assess the full extent of molecular convergence in $C_3$ to CCMs transitions has yet to be presented. For example, analyses of convergent evolution should include null hypotheses that assume no differences between taxa with and without convergence. In the case of CCMs evolution, a plausible null hypothesis consists in statistically equivalent numbers of convergent replacements between $C_4$ (or CAM) lineages and $C_3$ lineages.

Additionally, nonadaptive replacements should be used to normalize convergent replacements, in order to account for variation in the rates of nonsynonymous substitutions across lineages. This approach has been successfully applied in studies of molecular convergent evolution in vertebrates by assessing both convergent replacements and protein sequence changes that result in different amino acids, or *divergent replacements* (*Castoe et al., 2009*; *Thomas & Hahn, 2015*; *Zou & Zhang, 2015*). A broader definition of the latter group incorporates all replacements leading to different amino acids, regardless of their ancestral state. We refer to such changes as *non-convergent replacements*.

Furthermore, testing hypotheses about the extent of convergent molecular evolution remains particularly challenging for many nuclear genes, because of the prevalence of duplicated copies, particularly in plants (*Christin et al., 2007*; *Goolsby et al., 2018*). Single-copy nuclear or organelle genes allow to more easily recognize convergent changes and overcome possible confounding compensatory effects due to the presence of paralogous copies.

Given these premises, we sought to test if convergent amino acid changes occur more frequently in proteins encoded by chloroplast genes in a taxon that includes multiple well-characterized lineages of $C_4$ and $C_3$ grasses. Chloroplast proteins represent an ideal set of targets to study the role of convergent evolution in $C_3$ to $C_4$ transitions for a variety of reasons. First, most chloroplast proteins are involved in biochemical and biophysical processes that are critical to photosynthesis. For instance, out of ∼75 functionally annotated protein-coding genes in the maize chloroplast genome, 45 genes are implicated in photosynthesis-related processes, including *rbcL*, 17 genes coding for subunits

of the photosystems I and II (PS I and PS II), 12 genes coding for subunits of the NADH dehydrogenase complex, 6 genes coding for chloroplast ATPase subunits, 4 genes coding for cytochrome b6f complex subunits, and a few more genes implicated in the assembly of other protein complexes (*Maier et al., 1995*). Second, nonannotated orthologous copies of chloroplast genes can be readily identified across plants through sequence homology searches, taking advantage of the thousands of complete chloroplast genome sequences currently available for green plants. Third, comparative studies of convergent evolution in $C_4$ photosynthesis are facilitated by detailed reconstruction of phylogenetic relationships within groups with both $C_4$ and $C_3$ lineages. Fourth, signatures of positive selection have been found in multiple chloroplast genes in taxa that contain both $C_3$ and $C_4$ plants, although only the genes *rbcL* and *psaJ*, which encodes a small subunit of the Photosystem I complex, showed evidence of adaptive changes exclusively in $C_4$ lineages (*Christin et al., 2008*; *Goolsby et al., 2018*; *Piot et al., 2018*). Finally, most chloroplast genes occur as single copy loci, as opposed to the multiple paralogs typically present for plant genes encoded in the nucleus.

In this study, we analyzed 67 chloroplast genes from 64 grass species, including 43 $C_4$ and 19 $C_3$ species belonging to the PACMAD clade, named after six of its most representative subfamilies: Panicoideae, Arundinoideae, Chloridoideae, Micrairoideae, Aristidoideae and Danthonioideae. Using published information, we placed thirteen known independent $C_3$ to $C_4$ transitions in the reconstructed phylogeny of these 64 species. We applied a series of tests based on convergent *vs.* non-convergent amino acid replacements and determined that convergent molecular evolution occurred at a higher rate in chloroplast genes of $C_4$ lineages compared to $C_3$ lineages, a pattern that remained largely unchanged after excluding the RbcL protein from the convergence analyses. Our findings suggest that the evolutionary trajectories of multiple chloroplast genes have been affected during the emergence of the $C_4$ adaptation in the PACMAD clade, a result that has significant implications for our understanding of $C_4$ photosynthesis evolution.

## METHODS

### Data source and filtering

We queried NCBI GenBank (*Sayers et al., 2019*) for complete chloroplast genome sequences of grass species that were included in phylogenetic analyses by the *Grass Phylogeny Working Group II (2012)* and downloaded the corresponding coding sequences. Each species was assigned to either $C_3$ or $C_4$ type following the results of the *Grass Phylogeny Working Group II (2012)*. Additionally, we downloaded the coding chloroplast sequences for *Dichanthelium acuminatum*, *Thyridolepis xerophila*, *Sartidia dewinteri* and *Sartidia perrieri* ($C_3$ species) (*Brown & Smith, 1972*; *Smith & Brown, 1973*; *Hattersley & Stone, 1986*; *Hattersley et al., 1986*; *Besnard et al., 2014*). We used the standalone blastn ver. 2.2.29+ (*Camacho et al., 2009*) with the Expect value (E) cutoff of 1e-$^{10}$ to determine putative sequence orthology with coding sequences of the *Zea mays* chloroplast genes (*Maier et al., 1995*). Single copy putative orthologs that were present in more than 95% of the species were retained for further analysis (Table S1).

## Multiple sequence alignment

We aligned the individual sequences using TranslatorX ver. 1.1 (*Abascal et al., 2010*) and the multiple sequence aligner MUSCLE with default parameters. Alignments were further adjusted manually using BioEdit ver. 7.0.9.0 (*Hall, 1999*). Stop codons and sites that could not be aligned unambiguously were removed.

## Phylogeny reconstruction

We concatenated the individual sequence alignments and extracted third codon position sites for phylogeny reconstruction. We ran PartitionFinder ver. 1.1.1 (*Lanfear et al., 2012*) to identify the best partitioning scheme (partitioning by gene) for the downstream analysis using both Akaike information criterion (AIC) (*Akaike, 1973*) and Bayesian information criterion (BIC) (*Schwarz, 1978*). We then used maximum likelihood framework as implemented in RAxML ver. 8.2.10 (*Stamatakis, 2014*) to reconstruct the phylogeny. Branch support was estimated using 1,000 bootstrap replicates. *Oryza sativa* and *Brachypodium distachyon* from the BOP (Bambusoideae, Oryzoideae and Pooideae) clade were used as outgroup, whereas all ingroup species belonged to the PACMAD clade. We used FigTree ver. 1.4.0 (*Rambaut, 2012*) to rearrange and visualize the phylogeny, and the figures were edited further to improve readability and to indicate $C_4$/$C_3$ classification.

## Ancestral state reconstruction

We reconstructed ancestral states at each phylogenetic node for each individual gene using the program codeml from the software package PAML ver. 4.9a (*Yang, 2007*) and the basic codon substitution model (model = 0, NSsites = 0). The guide tree consisted of the cladogram of all species with available sequences for each individual gene. Sites with gaps in one or more PACMAD species were excluded.

## Definition and characteristics of "reference branches"

In the reconstructed PACMAD phylogeny, we identified the branches including the most recent common ancestors of $C_4$-only clades and $C_3$-only clades. We refer to these branches as "$C_4$ reference branches" and "$C_3$ reference branches", respectively (see Figs. 1 and 2). We then compared the inferred protein sequence of each reference branch with the inferred sequence in their ancestral branch (next branch toward the root), in order to identify individual site changes that occurred along reference branches.

To assess the number of convergent and non-convergent replacements, amino acid changes were compared in all possible pairs of reference branches. Replacements in two reference branches that resulted in the same state (amino acid) at a given site were considered convergent, regardless of whether the corresponding ancestral states were the same or different (*Castoe et al., 2009*). After identifying convergent replacements, we separated them into parallel and convergent changes (*Zhang & Kumar, 1997*; *Storz, 2016*). Likewise, two replacements were considered non-convergent if states at the descendant orthologous sites were different, regardless of the corresponding ancestral states (*Castoe et al., 2009*).

The pairwise comparisons between reference branches are akin to the phylogenetically independent contrast (PIC) method developed by Felsenstein (*Felsenstein, 1985*). In the

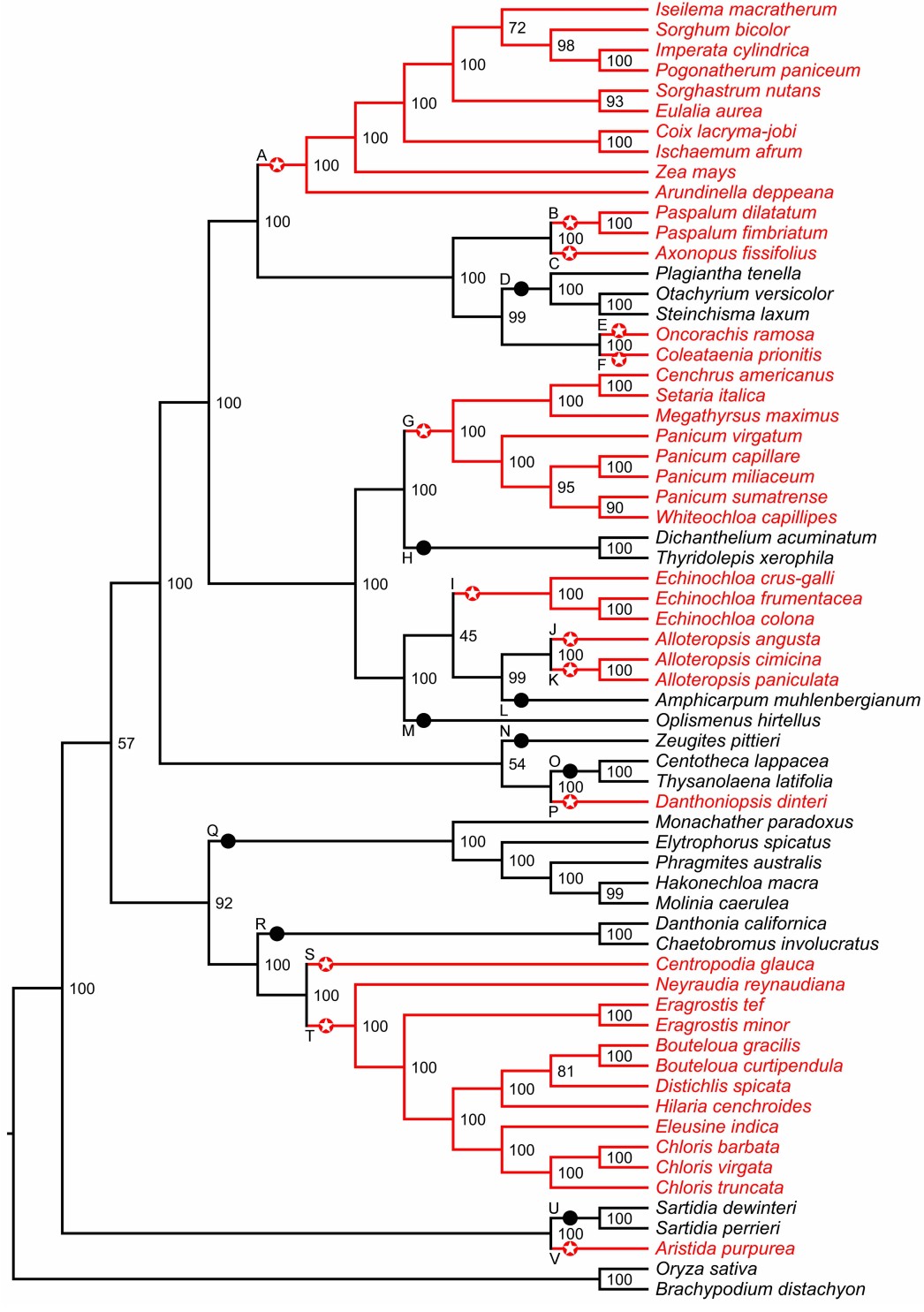

**Figure 1** **Phylogenetic relationships among 64 C₄ and C₃ grass species.** The phylogeny tree was obtained using RAxML (GTR+ Γ model) based on the third codon position sites in 67 chloroplast genes. The partitioning scheme was selected according to Akaike information criterion (AIC). C₄ and C₃ reference branches are shown in red and black, respectively. Red stars and black circles (labels A–V) indicate C₄ and C₃ reference branches, respectively. Numbers represent bootstrap support.

(A)                                                              (B)

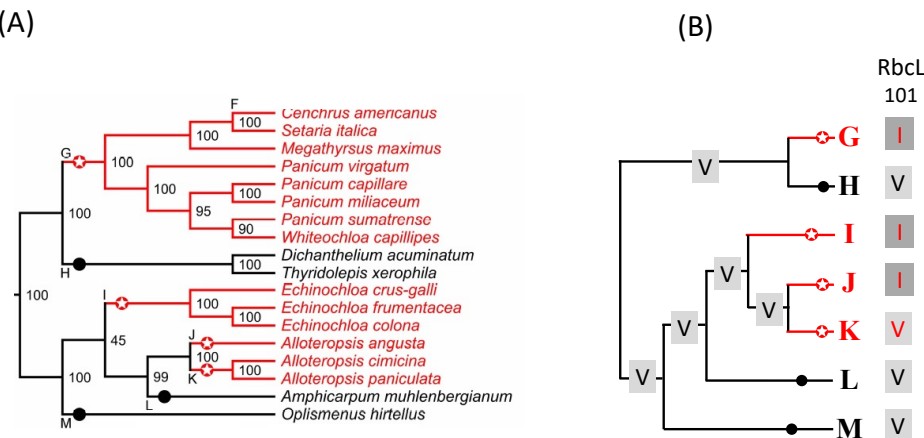

Phylogenetic reconstruction                 Replacements in ancestral branches

**Figure 2** **Example of $C_4$ and $C_3$ reference branches and convergent changes in $C_4$ reference branches.**
(A) PACMAD phylogeny and identification of reference branches. The $C_4$ reference branches (highlighted by red circles with stars) contain the common ancestor of a clade with only $C_4$ species (red lines). The $C_3$ reference branches (highlighted by black circles) contain the common ancestor of a clade with only $C_4$ species (black lines). $C_4$ reference branches that are next to each other represent lineages that independently acquired the $C_4$ pathway and are separated by species with the $C_3$ pathway that were no included in this study because of the lack of complete chloroplast genomes. For each species, the $C_4$ or $C_3$ photosynthesis type was obtained from the Supplementary figure 1 in the *Grass Phylogeny Working Group II (2012)*.
(B) Amino acid replacements in the reference branches. The sequence of chloroplast proteins was inferred in each reference branch and compared to the inferred sequence in the branch ancestral to the reference branch. In this example, the amino acid 101 in the protein RbcL is represented by a Valine (V) in branches ancestral to all reference branches, but a convergent V->I amino acid replacement occurred along the $C_4$ reference branches G, I and J.

PIC approach, the values to compare are represented by differences between branches. The differences between two branches are independent of the differences between two other branches. Therefore, pairwise comparisons of these values are independent and can be tested using $2 \times 2$ contingency table tests (see also below). In our study, pairwise comparisons are independent from each other, *i.e.,* replacements in each pair of branches are independent of replacements in each other pairs of branches. The difference from the PIC method is that we compare both differences (non-convergent replacements) and similarities (convergent replacements). A similar approach have been used in studies of convergent amino acid replacements (*Castoe et al., 2009*; *Foote et al., 2015*; *Thomas & Hahn, 2015*).

Reference branch lengths were extracted from the RAxML phylogeny obtained on the AIC partitioning scheme (Fig. S2). Testing was performed on the sum of pairs of branch lengths for each photosynthesis type using the R package exactRankTests (Table S2).

## Inference of convergent and non-convergent replacements and statistical testing

Using the approach described above, we identified putative convergent and non-convergent amino acid changes in each gene product individually. We summarized those data within

each of the three categories: (1) two $C_4$ reference branches ($C_4$:$C_4$), (2) $C_3$ reference branch and $C_4$ reference branch ($C_3$:$C_4$), and (3) two $C_3$ reference branches ($C_3$:$C_3$).

To test the significance of replacement differences between categories we used the Boschloo's exact unconditional test (*Boschloo, 1970*) implemented in the SciPy library ver. 1.7.1 in python3 (*Virtanen et al., 2020*). In the Boschloo's test, the *p*-value from the Fisher's exact test represents the test statistic of the exact unconditional test. It has been shown that Boschloo's test is more powerful than Fisher's exact test (*Mehrotra, Chan & Berger, 2003*). There is no restriction to using contingency table tests, including Boschloo's test, on categories with different sample size, as long as the categories are independent (*Mehrotra, Chan & Berger, 2003*), as in the case of reference branches in our phylogeny.

### Data availability
Raw data, including alignments, fasta sequences, and phylogenetic analyses data, are available through the following Figshare repository: https://figshare.com/articles/dataset/Convergence-chlorplast-genes-C4-Casola-Li-2021/15180690.

## RESULTS

### Phylogeny reconstructions
We examined 63 grass chloroplast genomes to identify gene orthologs for *Zea mays* chloroplast genes and extracted the corresponding coding and protein sequences. The resulting dataset included up to 67 DNA/protein sequences in 64 grass species that were retained for further analysis (Table S1). One to four sequences were absent in thirteen species. Out of 64 species, 43 were classified as $C_4$ and 21 (including two outroup species) as $C_3$. The reconstructed phylogeny is well supported, except for three branches with low to moderate bootstrap values, and it is consistent for both AIC and BIC partitioning schemes (Fig. 1 and Figs. S1–S3). We identified thirteen $C_4$ reference branches that represent putative $C_3$ to $C_4$ transitions, and nine $C_3$ reference branches (Fig. 1). Four pairs of reference branches corresponding to $C_3$ to $C_4$ transitions—B–C, E–F, J–L and S–T—are sister to each other in Fig. 1. Phylogenetic inferences from deep-taxonomic sampling of the PACMAD clade has shown that each of the these four pairs of reference branches is separated by at least one clade of $C_3$ species (*Grass Phylogeny Working Group II, 2012*), supporting the independent origin of $C_4$ photosynthesis in all reference branches shown in Fig. 1. However, no high-quality chloroplast genomes are available for any of the $C_3$ species between these pairs of reference $C_4$ branches, precluding their inclusion in our study.

Overall, the reference branches A-V showed support values that were in close agreement with those reported in the Grass Phylogeny Working Group II (2012), including the three branches with low statistical support in our tree. Importantly, the species topology was identical between the two phylogenies downstream these three branches. We also noticed three other branches that shared higher statistical support in our phylogeny compared to the Grass Phylogeny Working Group II tree. Two of these branches occurred in the subtribe Boivinellinae and correspond to the split between the group J/K and the branch L, and the split between the group I/J/K/L and the branch M (Fig. 1). The third node with higher support in our phylogeny correspond to the reference branch Q (tribe Arundoideae).

## Convergent and non-convergent amino acid replacements across chloroplast proteins

We assessed the level of molecular convergence in $C_3$ to $C_4$ transitions by quantifying convergent and non-convergent amino acid replacements across the PACMAD phylogeny by performing pairwise comparisons of reconstructed sequences in reference branches (Figs. 2 and 3, Table S3; see Methods). A total of 217 sites showed at least one convergent replacement: 104 in $C_4$:$C_4$, 120 in $C_3$:$C_4$ and 34 in $C_3$:$C_3$ pairs. A further 201 sites exhibited one or more non-convergent replacements: 96 in $C_4$:$C_4$, 121 in $C_3$:$C_4$, and 39 in $C_3$:$C_3$ pairs (Table 1). The difference in convergent/non-convergent site distributions between the three photosynthesis types was not statistically significant ($P \geq 0.05$, Boschloo's test; Table 1). The vast majority of convergent replacements shared the same ancestral state and should thus be considered parallel replacements according to widely accepted definitions of convergence (Zhang & Kumar, 1997; Storz, 2016). Only two sites, one in MatK (T205S/K205S in two $C_4$ reference branches) and the other in NdhF (L636I/K636I in one $C_4$ and three $C_3$ reference branches), shared replacements with different ancestral states, representing true convergent sites (Table S3).

To control for possible biases in the counting of convergent replacements due to branch length variation, we tested whether reference branch lengths in the three photosynthesis types $C_4$:$C_4$, $C_3$:$C_4$ and $C_3$:$C_3$ were different (Table S2). We found no significant difference among types ($P > 0.5$ for each of the three pairwise comparisons, Mann–Whitney $U$ test). We performed the same test only on branches with convergent and non-convergent replacements and found no significant difference between categories ($P > 0.5$, Mann–Whitney $U$ test; Table S2). Therefore, branch length variation between the three types is not expected to affect our results.

Among the $C_4$ reference branches, several individual sites showed high contrast in the number of branches involved in convergent and non-convergent replacements (Fig. 3, Tables S3 and S4). For example, seven $C_4$ branches (54%) shared the H18Q replacement in the product of *ndhH*, with no non-convergent replacements. Six, five, and four $C_4$ branches (46%, 38%, and 31%) showed convergent replacements at three sites in the RbcL protein (V101I, M309I, and A328S, respectively). Furthermore, six $C_4$ branches shared the S25G replacement in the product of *ndhI* and four L204F changes in the protein encoded by *matK*. In all these cases, there were no other convergent or non-convergent replacements in $C_3$:$C_3$ or $C_3$:$C_4$ branch comparisons, except for one H18Q change in NdhH in a $C_3$:$C_3$ branch. Two sites with convergent replacements in the proteins encoded by *ndhF* (L557F) and *rpoC2* (H875Y) were found uniquely in $C_3$:$C_3$ pairs, and only one site in the protein Rps3 showed convergence independently in $C_4$:$C_4$ and $C_3$:$C_3$ pairs (Fig. 3).

We then searched for convergent replacements that occurred along more than two $C_4$ branches at sites that remained otherwise conserved in $C_3$ and $C_4$ lineages, arguing that such changes could result from selective pressure rather than drift. We identified twelve $C_4$-specific convergent sites in proteins from 7 genes: *matK, ndhF, ndhG, ndhI, rbcL, rpoC1* and *rpoC2* (Table S4). Five of these sites were found in RbcL, whereas two sites were identified in NdhI. We also observed two convergent sites NdhF and one in RpoC2 that were uniquely found in three $C_3$ branches.

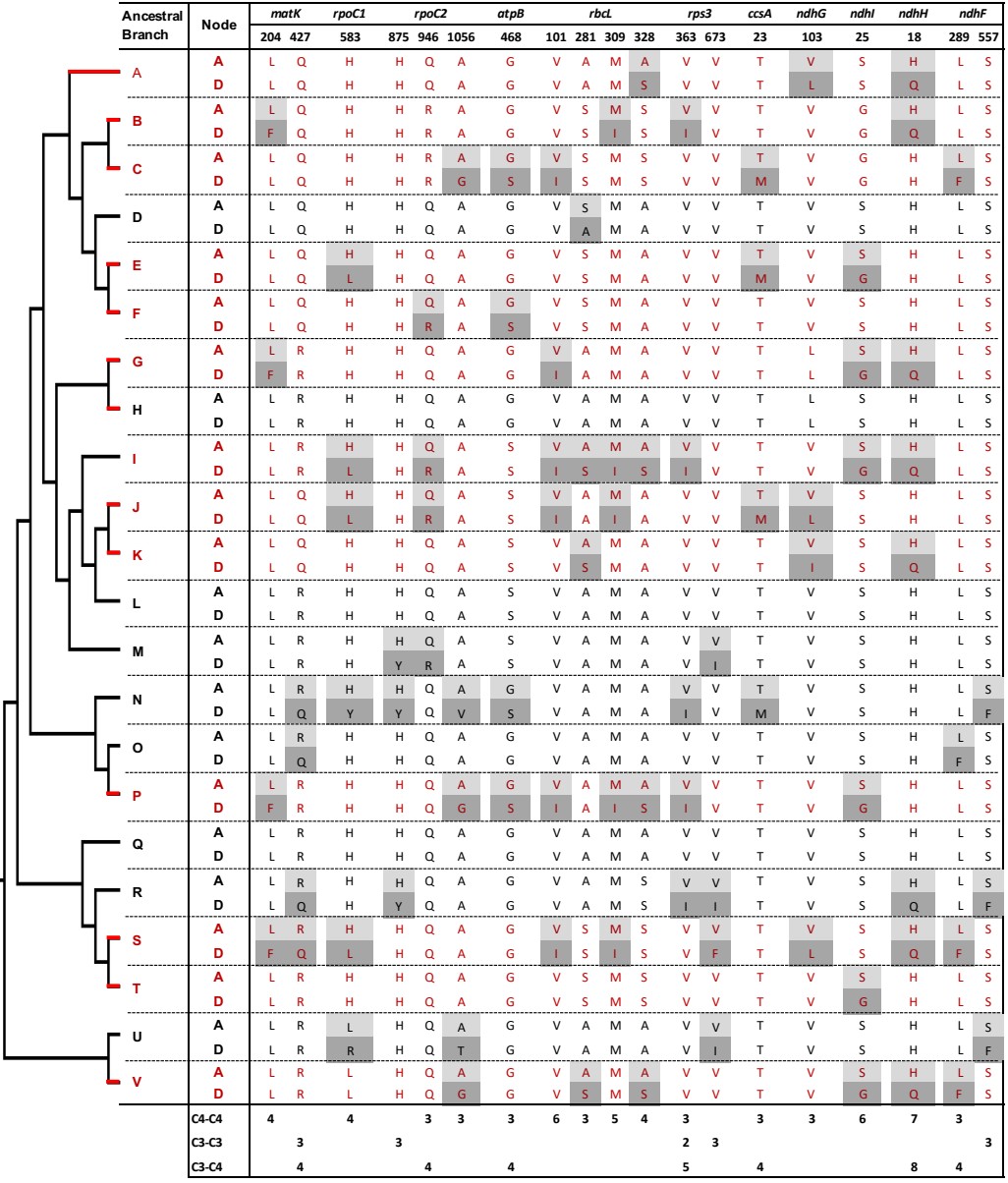

**Figure 3 Amino acid replacements shared by at least three C₄ or C₃ reference branches.** Ancestral (A) and derived (D) amino acids at replacement sites are shown. Site numbers correspond to the *Zea mays* orthologous sequence annotation. Red and black letters and branches represent C₄ and C₃ reference branches, respectively (see also Figs. 1 and 2).

## Molecular convergence in individual chloroplast proteins

Convergent and non-convergent amino acid replacements were detected in the products of 45 chloroplast genes, thirteen of which had at least one site with four or more replacements (Fig. 4, Tables 1 and S3). Twenty-four genes had convergent changes in C₄:C₄, 26 in C₃:C₄, and 13 in C₃:C₃ types of pairs (Table 1). Although the convergent/ non-convergent replacement ratio was higher in C₄:C₄ pairs than C₃:C₄ and C₃:C₃ pairs, the differences

**Table 1  Numbers of amino acid sites and genes with convergent and non-convergent replacements in reference branch comparisons.** Comparisons were made between pairs of $C_4$:$C_4$, $C_3$:$C_3$ and $C_3$:$C_4$ branches. Numbers of replacements unique to a given category (⋆), and the corresponding ratios Con:NC (Ratio). Differences between the $C_3$:$C_3$ and $C_4$:$C_4$ categories are not statistically significant ($P \geq 0.05$, Boschloos test).

| | $C_4$:$C_4$ | | | $C_3$:$C_4$ | | | $C_3$:$C_3$ | | |
|---|---|---|---|---|---|---|---|---|---|
| | **Con** | **NC** | **Ratio** | **Con** | **NC** | **Ratio** | **Con** | **NC** | **Ratio** |
| Sites | 104 | 96 | 1.08 | 120 | 121 | 0.99 | 34 | 39 | 0.87 |
| Sites⋆ | 80 | 64 | 1.25 | 82 | 69 | 1.19 | 17 | 16 | 1.06 |
| Genes | 24 | 23 | 1.04 | 26 | 32 | 0.81 | 13 | 17 | 0.76 |
| Genes⋆ | 24 | 20 | 1.2 | 25 | 29 | 0.86 | 9 | 10 | 0.9 |

**Notes.**
Con, convergent; NC, non-convergent.

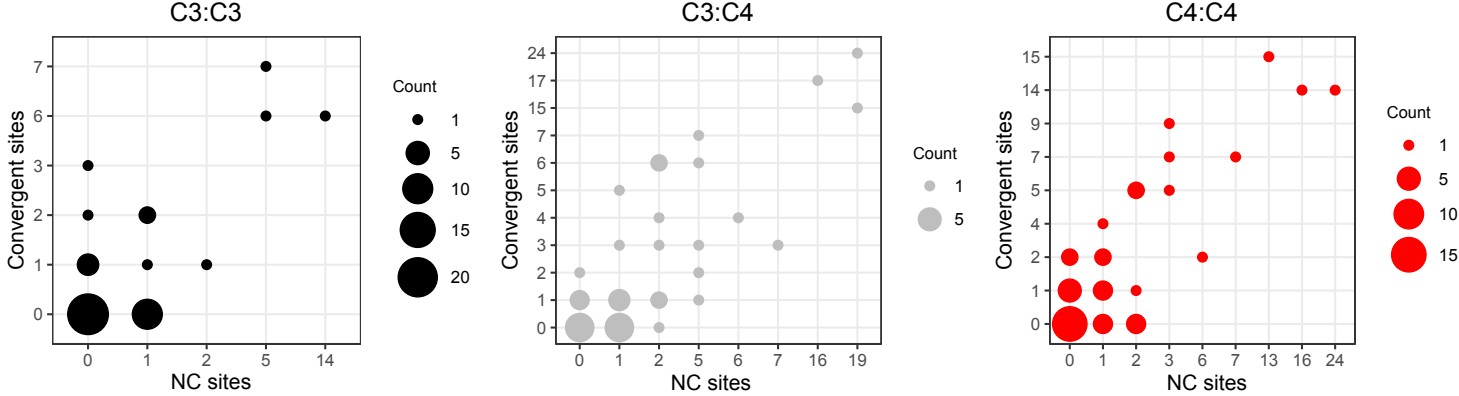

**Figure 4  Distribution of convergent and non-convergent amino acid replacements in pairs of reference branches.** (A) $C_4$:$C_4$ pairs. (B) $C_3$:$C_3$ pairs. (C) $C_3$:$C_4$ pairs. NC: non-convergent.

between the three photosynthesis types were not statistically significant ($P \geq 0.05$, Boschloo's test; Table 1). The lack of replacements was the single most common state for chloroplast proteins across photosynthesis types; however, in $C_4$:$C_4$ there were more genes with a higher number of convergent vs. non-convergent replacements (Fig. 4 and Table S5).

Overall, 26 proteins showed a higher number of convergent vs. non-convergent sites, of which 16, 13 and 10 were found in $C_4$:$C_4$, $C_3$:$C_4$ and $C_3$:$C_3$ pairs, respectively (Fig. 5 and Table S5). We found statistically significant differences in the number of convergent vs. non-convergent replacements between $C_4$:$C_4$ and $C_3$:$C_4$ pairs, but not $C_3$:$C_3$ pairs, in the products of the genes *rbcL*, *rpoC1* and *rpoC2* ($P < 0.05$, Boschloo's test; Table S5). In RbcL and RpoC1, $C_4$:$C_4$ pairs shared much higher proportion of convergent vs. non-convergent replacements, whereas the opposite was true in RpoC2. RpoC1 was also the only protein showing more convergent than non-convergent replacements in $C_4$:$C_4$ pairs compared to $C_3$:$C_3$ and $C_3$:$C_4$ pairs. In $C_4$:$C_4$ pairs, RpoC1 shared 4 convergent and 1 non-convergent replacement, compared to 1 and 2 in $C_3$:$C_3$ pairs and 1 and 5 in $C_3$:$C_4$ pairs, respectively. Additionally, the proteins NdhG, NdhI, PsaI, RpoA, Rps4 and Rps11 exhibited convergent

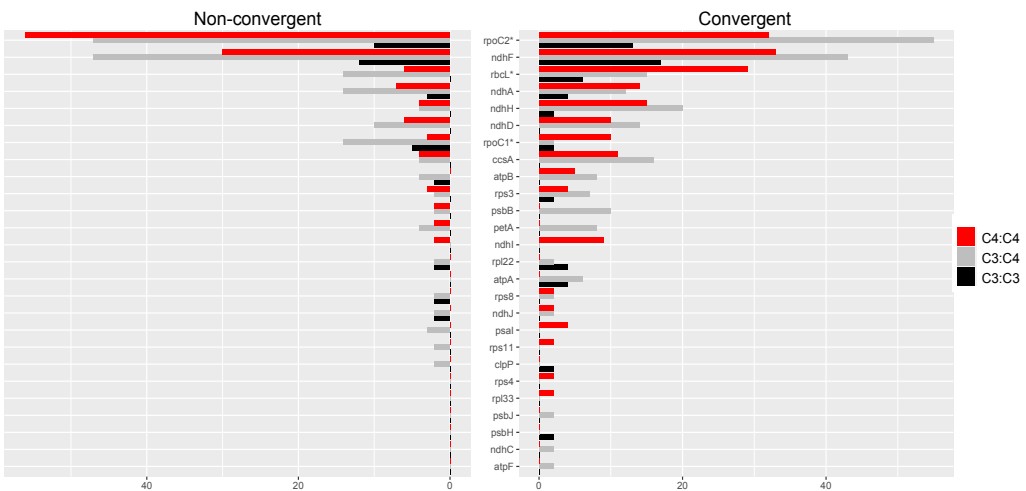

**Figure 5 Amino acid replacements in chloroplast proteins with more convergent than non-convergent changes.** Twenty-six chloroplast proteins with more convergent than non-convergent changes in $C_4$:$C_4$, $C_3$:$C_4$ and $C_3$:$C_3$ pairs.

replacements only in $C_4$:$C_4$ pairs (Table S5). When considering the number of affected sites rather than the number of replacements, no genes showed a significantly different pattern between photosynthesis types ($P \geq 0.05$, Boschloo's test; Table S5).

The proteins encoded by *matK*, *rpoC2* and *ndhF* shared much higher numbers of both convergent and non-convergent replacements than other chloroplast proteins across all photosynthesis type comparisons (Table S5). Both *matK* and *ndhF* are known to be rapidly evolving and have been consistently used in low taxonomic level phylogenetic studies in flowering plants (*Patterson & Givnish, 2002*; *Barthet & Hilu, 2008*). The gene *rpoC2* has also been recently described as a useful phylogenetic marker in angiosperms (*Walker et al., 2019*).

## Molecular convergence across reference branches

The comparison of reference branch pairs with convergent and non-convergent replacements revealed remarkable differences between photosynthesis types. Overall, $C_4$:$C_4$ pairs of reference branches showed a distribution skewed toward more convergent and non-convergent replacements than the two other categories (Fig. 6). There were significantly fewer pairs of $C_4$:$C_4$ reference branches with no replacements and with no convergent replacements than $C_3$:$C_4$ and $C_3$:$C_3$ pairs ($P < 0.05$, Boschloo's test; Table 2). Conversely, significantly more $C_4$:$C_4$ pairs shared more convergent than non-convergent replacements, and at least two convergent changes compared to $C_3$:$C_4$ and $C_3$:$C_3$ pairs ($P < 0.05$, Boschloo's test; Table 2). No significant difference was observed between pairs of $C_3$:$C_4$ and pairs of $C_3$:$C_3$. We found identical patterns when the same analyses were performed after excluding all replacements in the RbcL protein, except for the lack of
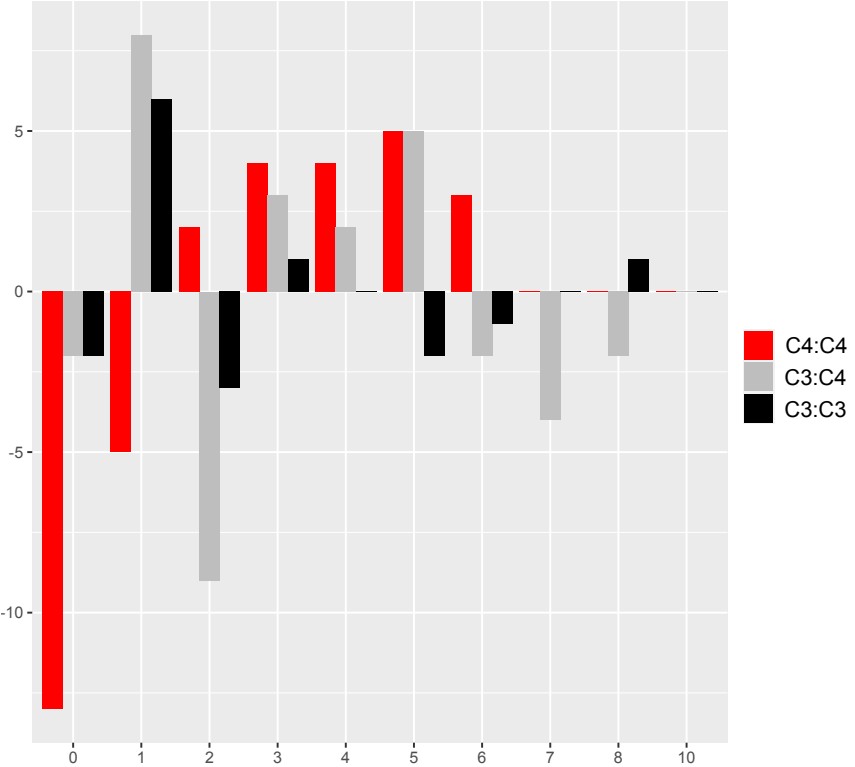

**Figure 6** **Pairs of reference branches by convergent and non-convergent replacements.** Difference in the number of pairs of reference branches for convergent and non-convergent categories (0–8 and 10 replacements).

a significant difference between $C_4$:$C_4$ and $C_3$:$C_3$ in the proportion of pairs with non-convergent replacements and pairs with more convergent than non-convergent changes (Table S6).

## Distribution of amino acid replacements across PACMAD lineages

Convergent and non-convergent replacements were preferentially found in specific pairs of reference branches. In $C_4$ pairs, convergent sites were most abundant between *Danthoniopsis dinteri* and *Aristida purpurea* (ten sites, branches P and V in Fig. 1), whereas non-convergent sites were most common between *Centropodia glauca* and *Aristida purpurea* (ten sites, branches S and V in Fig. 1). In pairwise $C_3$ branch comparisons, most convergent sites were identified between both *Zeugites pittieri* and Danthonieae (branches N and R in Fig. 1) and Danthonieae and *Sartidia* spp. (branches R and U in Fig. 1), whereas the most non-convergent site-rich pair was formed by *Zeugites pittieri* and *Sartidia* spp. (eight sites, branches N and U in Fig. 1; Table S7).

## Molecular convergence in the RuBisCO large subunit

We further inspected the evolution of the RuBisCO large subunit across the PACMAD clade. A total of 4 out of 9 RbcL amino acids with convergent changes in $C_4$ reference branches—V101I, A281S, M309I and A328S—have been identified in previous studies on

**Table 2  Number of reference branches with convergent and non-convergent replacements.** Comparisons were made between pairs of $C_4$:$C_4$, $C_3$:$C_3$ and $C_3$:$C_4$ branches. Proportions of pairs of reference branches over all branches by category are shown in parenthesis. The total number of pairs of reference branches are 78, 36 and 117 for $C_4$:$C_4$, $C_3$:$C_3$ and $C_3$:$C_4$ comparisons, respectively. All comparisons between $C_4$:$C_4$ pairs and both $C_3$:$C_3$ and $C_3$:$C_4$ pairs were statistically significantly different ($P < 0.05$, Boschloos test). No comparison between $C_3$:$C_3$ and $C_3$:$C_4$ pairs was statistically significant ($P \geq 0.05$, Boschloos test).

| | $C_4$:$C_4$ | $C_3$:$C_4$ | $C_3$:$C_3$ |
|---|---|---|---|
| No replacements | 6 (.08) | 30 (.26) | 12 (.33) |
| No Con | 12 (.15) | 48 (.41) | 16 (.44) |
| w/Con | 66 (.85) | 69 (.59) | 20 (.56) |
| w/NC | 63 (.81) | 67 (.57) | 18 (.50) |
| Con > NC | 40 (.51) | 36 (.31) | 10 (.28) |
| Con > 1 | 49 (.63) | 39 (.33) | 8 (.22) |

**Notes.**

Con, convergent; NC, non-convergent; Con > NC, pairs of branches with more convergent than non-convergent replacements; Con > 1, pairs of branches with more than one convergent replacement.

PACMAD grasses (*Christin et al., 2008*; *Piot et al., 2018*) as sites that experienced adaptive evolution in $C_4$ species (Table 3). A further site, T143A, was found to evolve under positive selection in $C_3$ to $C_4$ transitions in monocots (*Studer et al., 2014*). Interestingly, an adaptive S143A replacement has also been detected in the gymnosperm *Podocarpus* (*Sen et al., 2011*). Three more sites with convergent replacements—at positions 93, 94 and 461—correspond to amino acids that were reported to evolve under positive selection in different groups of seed plants by *Kapralov & Filatov (2007)*. Thus, all of the *rbcL* codons that appear to have evolved convergently among the PACMAD $C_4$ lineages we have examined are also known to have experienced adaptive evolution in seed plants, but not all of them have been shown to evolve adaptively in $C_4$ grasses.

## DISCUSSION

The recurrent emergence of carbon-concentration mechanisms (CCMs) across multiple angiosperm clades in the past 35 million years represents one of the most striking examples of convergent evolution of a complex phenotypic trait (*Sage, Christin & Edwards, 2011*; *Heyduk et al., 2019*). Several investigations have shown that the phenotypic parallelism across $C_4$ lineages is to some extent mirrored by convergent changes in the sequence of proteins with key metabolic roles in the biochemistry of $C_4$ photosynthesis, both in monocots and eudicots (*Christin et al., 2007*; *Besnard et al., 2009*; *Christin et al. 2009a*; *Christin et al. 2009b*; *Kapralov et al., 2011*; *Goolsby et al., 2018*). Furthermore, biochemical analyses have determined that some of these changes reflect adaptive shifts, as in the case of the increased availability of $CO_2$ at the RuBisCO site (*Studer et al., 2014*). Substantial changes in several RuBisCO kinetic traits associated to $C_3$ to $C_4$ transitions have recently been described (*Bouvier et al., 2021*). Further evidence of changes in the selective pressure associated to the $C_3$ to $C_4$ transitions have emerged from the detection of several positively selected sites in multiple genes associated with photosynthetic processes (*Christin et al., 2008*; *Studer et al., 2014*; *Goolsby et al., 2018*; *Piot et al., 2018*). These and other discoveries

**Table 3** Summary of RbcL amino acid sites with signatures of convergent evolution or positive selection.

| Codon | Ancestral AA | Convergent change/p.s.s. | #Convergent a.b. |
|---|---|---|---|
| **10** | **S** | **G** | **2** |
| **93** | **E** | **D** | **2** |
| **94** | **A** | **P** | **2** |
| **101**[*†] | **V** | **I** | **6** |
| 142[*†] | P | Several | na |
| **143** | **T** | **A** | **3** |
| 145[*†] | S | A/V | na |
| 258[*] | R | K | na |
| 270[*] | L | I | na |
| **281**[*†] | **A** | **S** | **3** |
| 282[†] | H | Several | na |
| **309**[*†] | **M** | **I** | **5** |
| **328**[*†] | **A** | **S** | **4** |
| **461**[*] | **V** | **I** | **2** |
| 468[†] | E | D | na |
| 471[†] | E | Several | na |
| 476[†] | I | L/V | na |

Notes.

Ancestral AA, ancestral amino acid; Convergent change/p.s.s, derived amino acid in multiple C4 reference branches and positively selected sites from previous studies; #Convergent a.b., number of reference branches with convergent changes.

Bold indicates sites with convergent changes identified in this study.

[*] Positively selected sites in PACMAD C4 lineages from *Christin et al. (2008)*.

[†] Positively selected sites in PACMAD C4 lineages from *Piot et al. (2018)*.

have paved the way to a more nuanced understanding of the molecular basis of phenotypic convergence in CCM plants and may accelerate the development of crop varieties with augmented resistance to high temperature and low water availability.

For these aims to be fully realized, a robust framework to assess the extent and phenotypic impact of convergent molecular changes is necessary. Along the lines of strategies applied in vertebrates research (*Castoe et al., 2009*; *Foote et al., 2015*; *Thomas & Hahn, 2015*; *Zou & Zhang, 2015*), we presented here the results of a novel methodological approach to the study of molecular convergence in $C_4$ grasses. We investigated patterns of convergent and non-convergent amino acid changes in nearly 70 chloroplast proteins across multiple $C_4$ and $C_3$ lineages in the PACMAD clade, with the goal of testing a specific hypothesis: is the evolution of chloroplast proteins showing stronger signatures of convergent amino acid replacements in $C_4$ lineages compared to $C_3$ lineages? This analysis also allowed us to establish if proteins other than enzymes involved in the CCM biochemistry underwent parallel amino acid changes in $C_4$ lineages. Our reasoning is that many proteins expressed in the chloroplast could have experienced similar selective pressure across multiple $C_3$ to $C_4$ transitions and might have accumulated convergence replacements as a result. In agreement with our expectation, dozens of nuclear genes sharing signatures of positive or relaxed selection and likely associated with the evolution of $C_4$ PACMAD grasses have

been recently described, albeit these analyses relied on a limited number of species (*Huang et al., 2017*).

We based our analysis on the identification of amino acid replacements shared by pairs of reference $C_4$ branches, defined here as branches corresponding to $C_3$ to $C_4$ transitions in the PACMAD phylogeny. We compared these changes to those identified in reference $C_3$ branches, namely all $C_3$ lineages that include only $C_3$ species (Figs. 1 and 2), and to changes found between reference $C_3$ and $C_4$ branches. For each of the three possible pairs of photosynthesis types, $C_4$:$C_4$, $C_3$:$C_4$ and $C_3$:$C_3$, we determined the number of amino acid sites, genes and pairs of reference branches with convergent replacements.

We detected signatures of convergent evolution in all types of datasets. First, we identified many individual replacements that emerged repeatedly and uniquely in $C_4$ reference branches, particularly in the proteins RbcL, NdhH, NdhI and MatK. We also observed $C_3$-specific convergent replacements in NdhF and RpoC2, and a case of multiple $C_4$ and $C_3$ convergent changes in Rps3. Additionally, we identified 7 chloroplast genes with one or more $C_4$-specific convergent sites and 3 chloroplast genes with at least one $C_3$-specific convergent site. Second, we found evidence of significantly higher rates of convergent replacements in $C_4$ lineages in both RbcL and RpoC1, and several convergent replacements that occurred exclusively in $C_4$:$C_4$ pairs in proteins encoded by *ndhG*, *ndhI*, *psaI*, *rpoA*, *rps4* and *rps11*. These genes are involved in a variety of biological processes in the chloroplast, from the cyclic electron transport in (*ndhG* and *ndhI*) and the stabilization of (*psaI*) the photosystem I, to transcription (*rpoA* and *rpoC1*), translation (*rps4* and *rps11*) and $CO_2$ fixation (*rbcL*). Third, we identified statistically significant differences in pairs of $C_4$ branches with convergent replacements (Table 2). Crucially, we observed more pairs with higher convergent than non-convergent replacements in $C_4$:$C_4$ compared to both $C_3$:$C_3$ and $C_3$:$C_4$, even after removing replacements identified in the RuBisCO large subunit, RbcL.

Altogether, these findings suggest that multiple biochemical processes occurring in the chloroplast might have experienced recurrent adaptive changes associated with the emergence of $C_4$ photosynthesis. Notably, some of these proteins are not directly involved in the light-dependent or light-independent reactions of the photosynthesis, implying that processes such as regulation of gene expression and protein synthesis in the chloroplast are also experiencing significant selective pressures during the transition from $C_3$ to $C_4$ plants. These results should motivate further studies to determine the prevalence of convergent amino acid replacements in transitions to CCMs among the thousands of proteins encoded by nuclear genes but expressed in the chloroplast (*Jarvis & Lopez-Juez, 2013*). Although such analyses are currently hindered by the limited number of sequenced nuclear genomes in taxa with multiple $C_3$ and $C_4$ lineages, including the PACMAD clade, genome-wide investigations of convergent replacements will be possible in the near future given the current pace of DNA sequencing in plants.

A further important conclusion drawn from these results is that convergent replacements are not uncommon between $C_3$:$C_3$ and $C_3$:$C_4$ lineages. This is possibly due to some environmental factors affecting the evolution of chloroplast genes that are shared across grass lineages regardless of their photosynthesis type.

The analysis of individual convergent replacements in the RuBisCO large subunit both confirmed previous findings (*Christin et al., 2008*; *Studer et al., 2014*; *Piot et al., 2018*) and highlighted novel potentially adaptive changes among PACMAD species. Importantly, these novel convergent replacements are known to evolve under positive selection in non-PACMAD seed plants (*Kapralov & Filatov, 2007*; *Sen et al., 2011*). This underscores the potential of our approach to identify novel changes with functional significance in the transition to CCMs in grasses, as opposed to standard statistical tests of positive selection. Alternatively, some RbcL sites could experience convergence across a variety of seed plants because of selective pressure other than those associated with $C_3$ to $C_4$ transitions.

Overall, our results are robust to several possible confounding factors. First, we analyzed branches that are strongly supported in our phylogeny reconstruction. The phylogenetic tree built using the 67 chloroplast genes is well supported, with the exception of three branches with fairly low bootstrap support. However, all three branches are short and have minimal impact upon our conclusions regarding $C_4$ evolution (Fig. 1 and Figs. S1–S3). Moreover, the tree is largely consistent with a comprehensive recent study of 250 grasses based on complete plastome data (*Saarela et al., 2018*). Second, by focusing only on reference branches and ignoring amino acid replacements that may have occurred after the divergence of species within a given $C_4$ clade, our strategy provided a conservative estimate of the number of convergent changes that could have occurred during the evolution of PACMAD grasses. Third, we eliminated genes with possible paralogous copies, which could have introduced false positive replacements.

We recognize some potential caveats in our approach. By relying on a relatively small sample of PACMAD species, our statistical power to detect signatures of convergent evolution was limited. Increasing the number of reference $C_4$ and $C_3$ lineages should provide a broader representation of convergent replacements in $C_4$ clades. Furthermore, we applied a strict definition of convergence that ignores changes to amino acids with similar chemical properties. We think that a conservative approach was necessary given that amino acids with similar chemical properties might have a very different functional effect on protein activity given their size and tridimensional interactions with nearby residues. Third, we assumed that all the observed convergent replacements were the result of convergent phenotypic changes, which fall under the general category of homoplasy (*Avise & Robinson, 2008*). However, some of these replacements could instead represent hemiplasy, or character state changes due to introgression between different $C_4$ lineages, incomplete lineage sorting (ILS) of reference alleles or horizontal gene transfer (*Avise & Robinson, 2008*). Recombination between chloroplast genomes, which is required for introgression to occur, has been documented but appears to be rare (*Carbonell-Caballero et al., 2015*; *Greiner, Sobanski & Bock, 2015*; *Sancho et al., 2018*). Introgression or horizontal gene transfer between congeneric species has been associated to the acquisition of part of the $C_4$ biochemical pathway in the PACMAD genus *Alloteropsis* (*Christin et al., 2012*; *Olofsson et al., 2016*). However, these transfers were limited to a few nuclear genes. Moreover, only a very few cases of horizontal transfer between chloroplast genomes have been reported in plants (*Stegemann et al., 2012*). Therefore, the contribution of hemiplasy to the observed pattern of convergent replacements in $C_4$ lineages is likely to be minimal. Finally, we treated

$C_4$ species regardless of their photosynthesis subtype (NAPD-ME, NAD-ME and PEPCK), which is known to vary among PACMAD subfamilies (*Taylor et al., 2010*). We argue that our results are conservative with regard to this aspect because convergent replacements should be expected to occur more often between $C_4$ groups sharing the same photosynthesis subtype.

## CONCLUSIONS

In this study, we showed that molecular convergent evolution in the form of recurrent amino acid replacements affected multiple chloroplast proteins in $C_4$ lineages of the PACMAD clade of grasses. This finding significantly broadened the number of genes known to have evolved convergently in $C_4$ species. We observed for the first time that genes not directly involved in photosynthesis-related processes experienced convergent changes, suggesting that future efforts should rely whenever possible on genome-wide analyses of amino acid changes rather than focus primarily on candidate key metabolic genes, similarly to previous investigations on gene expression patterns in $C_4$ and CAM plants. Our methodological approach based on the comparison of convergent and non-convergent replacements among photosynthesis types underscores the importance of a more rigorous hypothesis-based testing of convergent evolution signatures in $C_4$ plant evolution. Our results should inform more nuanced approaches to introduce CCM-like processes in $C_3$ crops.

## ACKNOWLEDGEMENTS

We thank two reviewers for comments and suggestions that have led to a significant improved version of this manuscript, and to A. Michelle Lawing for comments on the manuscript.

### Funding

The project was supported by the National Institute of Food and Agriculture, U.S. Department of Agriculture, under award number TEX0-1-9599, the Texas A&M AgriLife Research, and the Texas A&M Forest Service. The funders had no role in study design, data collection and analysis, decision to publish, or preparation of the manuscript.

### Grant Disclosures

The following grant information was disclosed by the authors:
The National Institute of Food and Agriculture.
U.S. Department of Agriculture: TEX0-1-9599.
The Texas A&M AgriLife Research.
The Texas A&M Forest Service.

### Competing Interests

The authors declare there are no competing interests.

## Author Contributions

- Claudio Casola conceived and designed the experiments, analyzed the data, prepared figures and/or tables, authored or reviewed drafts of the paper, and approved the final draft.
- Jingjia Li analyzed the data, prepared figures and/or tables, and approved the final draft.

## Data Availability

The data is available at figshare: Casola, Claudio (2021): Convergence-chlorplast-genes-C4-Casola-Li-2021. figshare. Dataset. https://doi.org/10.6084/m9.figshare.15180690.v2.

## Supplemental Information

Supplemental information for this article can be found online at http://dx.doi.org/10.7717/peerj.12791#supplemental-information.

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
