# Peer review of "Beyond RuBisCO: convergent molecular evolution of multiple chloroplast genes in C4 plants"

_PeerJ, doi:10.7717/peerj.12791_

## Round 0.1 · original submission · Major Revisions

The paper is very interesting and written clearly and appropriately; however, some improvements are requested before it may be considered for publication in PeerJ.

I recommend that the author strictly follow reviewers' instructions and respond to the reviewer's revisions with a point-to-point rebuttal.

Best Regards

·

Basic reporting

This is a really interesting and important paper. The findings are fascinating and will be of interest to the C4/photosynthesis community. There are some issues regarding terminology that would benefit from further clarification (e.g. use of convergence/divergence is not formally correct) and there are some issues that need more detailed explanation (better definition of ancestral branches, pairs of branches etc.). An improved and more step by step workflow and analysis figure (like and expanded version of Figure 2) might help the readers to better understand.

The only major criticism I have is that the claims concerning differences in number of convergent/divergent changes between cannot be upheld. The data is phylogenetically structured and has differences in sampling and branch length between groups. These three factors confound and invalidate the statistical tests that are used. However, even if these claims are removed this is a fantastically interesting paper. It doesn't need the difference in number/rate claims to be really interesting to the community. The finding that there are convergent/parallel changes in multiple chloroplast genes when plants evolve C4 photosynthesis will be very thought provoking.

Other comments are provided below.

Abstract
terminology “pairs of branches” is not clear and is important to understand to enable the reader to understand the results.

Introduction
General point: The authors dont distinguish between convergence and parallelism. Moreover, throughout this article convergence is used to describe parallelism. The authors should probably tighten these descriptions up so that they match something like what is put forward in Stern 2013 Nature reveiws genetics 14:751. This would avoid some confusion.

Line 74: replace “the few slightly different variants of the C4 photosynthesis type originated” with “C4 photosynthesis evolved”

Line 85: It would be good to cite the Heckman model of C4 evolution here and separate the desrcipriton of the liekly order of events from the amino acid changes.

Line 93: change “initiates” to catalyses. Also remove “limited produciton of energy” as it doesnt produce energy per se.

Line 99-112: Its probably worth mentioning the high throughput study of positive selection from the brutnell group in this context. Particulalry given line 130-132. It sort of has been done for a very limited species set.

Line 107: Might be worth citing Bouvier et al 2021 MBE for evidence for convergent adaptation of ribisco kinetics in C4s.

Line 175: As written it sounds like you identified the origins/transitions to C4. I think these were already known.

Line 181: delete remarkably

Line 183: I dont think the statement regarding interactions is true. I do think it has implications for our understanding of photosynthetic adaptation in C4s. I dont think they will be critical for engineering C4 into C3 crops. They may be useful in finetunig C4 function, but not critical for its engineering… the C3 C4 hybrids in Moricandia, alloteropsis etc prove this.

Line 205: what flavour alignment for TranslatorX (MAFFT? If so what mafft method).

Line 223: more details needed for ancestral state estimation. Individual alignments? Which guide tree? With sites removed?

Line 230-237: I think this needs a dedicated figure. Its conceptually complicated to understand and not labelled in Figure 1 to a level that its sufficient to understand what the authors have done. In the tree there are multiple origins of C4 annotated to monophylletic clades (e.g. alloteropsis, or paspalum and axonopus). Why is this the case? Should they not be annotated as single origins of C4?

Line 239: I dont understand what a “pair of ancestral branches” is.

Line 240-241: That is the definition of parallelism and not convergence.

Line 244: I think this is a little unfair. If two different states evolve to two other non-identical states they are no more or less divergent than before and so that should not be counted as divergence. Non-convergent or non-parallel changes would be a good name for the category.

Line 246: Time steps and time stamps are not precise terminology here. Refer to most likely ancestral sequences.

Line 251-257: I know its a pain but the authors need to change their terminology C4-C4 C3-C4 etc as C3-C4 is a widely used defined term in the literature that means something else. Perhaps just replace them all with “C3 to C4” or equivalent.

Line 269-276: It would be good to mention if you phylogeny disagrees with the APG grass phylogeny. Yours is more data rich and more likely to be correct. But ti would be good to know what the differences were and where those differences are.

Line 275-277: I have difficulty reconciling the number of C4 origins claimed with the toplogy of the tree. There seems to be some over counting here.

Line 280-355: There is a small problem with stats here. Because there are more C4 species and a larger number of C4 origins there is an inherent higher probability of observing a C4 convergent/parallel change. (i.e. if the probability of observing a convergent change on a branch is the same then if you if you increase the number of branches being analysed you are more likely to observe a convergent change). This needs to be correctly accounted for in the statistics to validate the claims of a higher occurrence. The length of the branches also needs to be accounted for here if a claim re rate or frequency of occurrence is to be supported. i.e. if probability of a convergent change is a function of time, and branch length is a function of time then the longer the branch the more likely you are to observe a convergent change. I have no doubt that the convergent changes observed by the authors are real. Its really interesting and exciting. I just think the claims about frequency and rate are not supported as the number of taxa and their phylogenetic distance and position and the number of C4 origins have not been taken into consideration in the statistical tests. i.e. Boschloos test is not valid as the data is phylogeneticaly structured. Some sort of phylogenetic comparative method will need to be used here if claims concerning relative numbers of changes or frequency of changes between groups of species are to be upheld. It should be noted that even if the statistics prove challenging and the authors remove all of the comparative data and claims (i.e. C4s have more changes than C3) then this will still be a really nice paper as provides a really nice list of convergent changes in the evolution of C3 to C4.

Experimental design

See above

Validity of the findings

See above

Additional comments

See above

Reviewer 2 ·

Basic reporting

The study by Casola & Li presents the results from an empirical approach to test molecular convergence in C3 and C4 taxa testing if convergence occurred more frequently in C4 compared to C3 lineages using multiple chloroplast proteins beside the large RuBisCO subunit.
Overall, the manuscript is well written and easy to read. The structure of the article and the quality of the figures are good and the authors also provide several supplementary data to support their findings.
The Introduction section considers a sufficient literature to justify the study carried out by the authors, whereas authors must include more references in the Discussion section to discuss their findings. The methods used are valid and appropriate to provide results for testing the experimental hypotheses of the study.

Some concerns arise regarding the use of terms such as "ancestral branches" and "pairs of branches" that need to be better clarified in the text to improve the reading and understanding of the article.

MINOR EDITS:

Line 199: use superscript for "e-10"

Line 294: "Showed" must not be capitalized

Line 318: Correct as follows: "higher number OF convergent vs. divergent replacements"

Experimental design

The experimental design is adequate for the purposes of the study and has been endowed with a sufficient level of scientific novelty. Indeed, the comparative analysis performed in this study is satisfactory for evaluating convergent and divergent amino acid substitutions along the C3-C4 phylogeny.

Validity of the findings

Overall, the findings of this study are well supported. In addition, these findings can be of great interest in the understanding of C4 photosynthesis evolution, possibly promoting future research in this topic. However, I suggest to revise the assertions regarding the differences in convergent/divergent changes between groups considering possible confounding results due to the differences in branch length between groups included in this study.

Additional comments

No additional comments

---

## Round 0.2 · accepted · Accept

Dear Authors,
The reviewers and I have appreciated all your efforts in improving the first version of the manuscript. In this new version, all criticisms have been resolved.

I have the pleasure to communicate that your paper has been accepted for publication in Peer J.

Best Regards

·

Basic reporting

The authors did a great job in revising the manuscript. I'm not completely aligned with the way in which the data was analysed to account for phylogeny, however I accept the authors arguments and that their approach has validity. This should be published. its really interesting.

Experimental design

See above

Validity of the findings

See above

Additional comments

The authors did a great job in revising the manuscript. I'm not completely aligned with the way in which the data was analysed to account for phylogeny, however I accept the authors arguments and that their approach has validity. This should be published. its really interesting.

Reviewer 2 ·

Basic reporting

The authors satisfied the requested revisions.

Experimental design

The authors satisfied the requested revisions.

Validity of the findings

The authors satisfied the requested revisions.